# Outcome of Application of Cryopreserved Amniotic Membrane Grafts in the Treatment of Chronic Nonhealing Wounds of Different Origins in Polymorbid Patients: A Prospective Multicenter Study

**DOI:** 10.3390/bioengineering10080900

**Published:** 2023-07-29

**Authors:** Alzbeta Svobodova, Vojtech Horvath, Lukas Balogh, Martina Zemlickova, Radovan Fiala, Jan Burkert, Marek Brabec, Petr Stadler, Jaroslav Lindner, Jan Bednar, Katerina Jirsova

**Affiliations:** 12nd Department of Surgery—Department of Cardiovascular Surgery, First Faculty of Medicine, Charles University and General University Hospital in Prague, 128 08 Prague, Czech Republic; alzbeta.svo@seznam.cz (A.S.); jaroslav.lindner@vfn.cz (J.L.); 2Department of Vascular Surgery, Na Homolce Hospital, 150 30 Prague, Czech Republic; vojtech.horvath@homolka.cz (V.H.); petr.stadler@homolka.cz (P.S.); 3Laboratory of the Biology and Pathology of the Eye, Institute of Biology and Medical Genetics, First Faculty of Medicine, Charles University, 128 00 Prague, Czech Republic; lukas.balogh@lf1.cuni.cz (L.B.); jan.bednar@lf1.cuni.cz (J.B.); 4Clinic of Dermatovenerology, General Teaching Hospital and First Faculty of Medicine, Charles University, 121 08 Prague, Czech Republic; martina.zemlickova@vfn.cz; 5Department of Cardiovascular Surgery, Motol University Hospital, 150 06 Prague, Czech Republic; radovan.fiala@fnmotol.cz (R.F.); jan.burkert@fnmotol.cz (J.B.); 6Department of Transplantation and Tissue Bank, Motol University Hospital, 150 06 Prague, Czech Republic; 7Department of Statistical Modeling, Institute of Computer Science, The Czech Academy of Sciences, 182 07 Prague, Czech Republic; mbrabec@cs.cas.cz

**Keywords:** nonhealing wounds, cryopreserved amniotic membrane, polymorbid patients, pain

## Abstract

To compare the therapeutic efficacy of cryopreserved amniotic membrane (AM) grafts and standard of care (SOC) in treating nonhealing wounds (NHW) through a prospective multicenter clinical trial, 42 patients (76% polymorbid) with 54 nonhealing wounds of various etiologies (mainly venous) and an average baseline size of 20 cm^2^ were included. All patients were treated for at least 6 weeks in the center before they were involved in the study. In the SOC group, 29 patients (36 wounds) were treated. If the wound healed less than 20% of the baseline size after 6 weeks, the patient was transferred to the AM group (35 patients, 43 wounds). Weekly visits included an assessment of the patient’s condition, photo documentation, wound debridement, and dressing. Quality of life and the pain degree were subjectively reported by patients. After SOC, 7 wounds were healed completely, 1 defect partially, and 28 defects remained unhealed. AM application led to the complete closure of 24 wounds, partial healing occurred in 10, and 9 remained unhealed. The degree of pain and the quality of life improved significantly in all patients after AM application. This study demonstrates the effectiveness of cryopreserved AM grafts in the healing of NHW of polymorbid patients and associated pain reduction.

## 1. Introduction

Treatment of chronic nonhealing wounds is a serious health and socio-economic global problem. The increasing number of nonhealing wound occurrences is tightly related to the global aging of society and a high prevalence of age-related diseases. These factors increase the rate of wound formation and negatively affect their healing. Chronic nonhealing wounds seriously deteriorate the quality of life of affected subjects in the sense of elimination or discrimination from society, limited mobility, and productivity [1]. Industrialized countries spend 2–4% of the total healthcare budget on treating chronic wounds [2]; the average cost of wound care ranges from €6000–€10,000 in Europe [3] and reaches more than $70 billion yearly in the United States [2].

Due to the fact that chronic wound is not defined formally and unambiguously, the European Wound Management Association proposed using the term “nonhealing wounds” (NHW) [4], a term that will be used herein. For the wound to be designated as NHW, the time of its resistance to treatment has been defined in the range of four weeks up to more than three months [5,6,7]. Common features of NHW include repeating or persistent infections, inflammation, formation of drug-resistant biofilms, and loss of ability to respond to reparative stimuli [8,9].

The most common etiologies of NHW include chronic venous insufficiency (CVI), diabetes mellitus, peripheral artery disease (PAD), and post-traumatic conditions. Many patients with long-term NHW wounds are polymorbid (suffering from two or more chronic diseases) [10,11,12]. It was shown that multiple comorbidities negatively affect the success of complete wound closure [8]. Cost and time reduction and increased NHW treatment efficiency are the major aims of modern wound care. With the increasing size and duration of the nonhealing period of a wound, the requirements for more specific and effective therapy grow steeply.

Using biological materials or wound dressing with the presence of biologically active compounds has proven to be a very effective treatment [13,14]. Recent trends highlight a shift to biological or hybrid wound care systems. Body-related products naturally substitute extracellular matrix components, nutrients, and other factors necessary for healing in the wound area. Such treatment includes, for example, the use of artificial skin or placental derivatives. Recently, the use of even more specific regenerative approaches, consisting of the use of cell-based therapy, e.g., mesenchymal stem/stromal cells [15] or even cell parts or products such as extracellular vesicles (exosomes) or subcellular components have been introduced in clinical practice [16,17].

The methodological progress and growth of clinical experience with treating wounds using placental membranes over the past 20 years have shown the usefulness of alternative appropriate biological dressings in accelerating or inducing wound healing [18,19]. The presence of growth factors (e.g., epidermal growth factor, keratinocyte growth factor, hepatocyte growth factor, basic fibroblast growth factor, platelet-derived growth factor), cytokines (IL-2, IL-8, IL-10), protease inhibitors (tissue inhibitors of metalloproteinases, serpins), and other compounds (e.g., hyaluronic acid) ensure the promotion of epithelization and the anti-inflammatory and anti-fibrotic features of AM [20,21,22,23,24]. Currently, mainly cryopreserved [25,26] and dehydrated amniotic [25] or amniochorionic membranes [27] are often used clinically to treat NHW of different etiologies [28,29]. Besides accelerating the healing, amniotic and amniochorionic grafts exhibit significant analgesic effects after application to patients with burns [30] or skin defects [31,32].

In the present multicenter study, we assessed the effect of cryopreserved AM application on NHW of various etiologies (venous, arterial, postoperative, diabetic). However, prior to the AM application, most subjects were pretreated by intense and well-documented SOC in the centers for a minimum of 6 weeks to eliminate the possibility of inadequate SOC application in outpatient care. By evaluating the healing progress, we aimed to determine the efficacy of AM application and analyze whether the progress of wound closure can be used as a predictor/estimator for the efficacy of the AM treatment of NHW. Simultaneously we evaluated progress in pain relief and quality of life after AM application.

## 2. Materials and Methods

This multicenter prospective trial assessed the effect of the application of cryopreserved AM allografts compared to SOC alone on patients with NHW. It was approved by the Ethical Committees of three participating institutions (1st Faculty of Medicine Charles University, General University Hospital, University Hospital Motol, and Na Homolce Hospital, all in Prague) and adhered to the tenets set out in the Declaration of Helsinki.

### 2.1. AM Graft Preparation

Cryopreserved AM grafts were prepared as described previously [33]. Briefly, a graft containing an intact AM was aseptically processed from placentas donated by healthy screened mothers after a caesarian section (medical record and personal history was evaluated to prevent the transmission of genetic and infectious diseases). AM was decontaminated by antibiotic solution (Base 128, Alchimia, s.r.l., Ponte San Nicolò, Padova, Italy), then minimally processed (cleaned from blood clots) and cryopreserved (−80 °C) in a mixture of Dulbecco’s Modified Eagle’s Medium (DMEM, c.n. 32430-027, Gibco Life Technologies, Invitrogen, Waltham, MA, USA) and glycerol (Glycerolum 85%, Dr. Kulich Pharma s.r.o., Hradec Kralove, Czech Republic) 1:1 [34]. During tissue processing and packaging, sterility tests were performed. Only allografts with negative serology (HIV, hepatitis B, and C, syphilis), both on the day of tissue retrieval and 180 days after, were released for grafting.

### 2.2. Subjects Enrolment, Study Groups

The patients were recruited following the inclusion criteria: age ≥ 18 years, resistant NHW with a duration of more than 6 weeks, and wound of a maximum size of 100 cm^2^ extending through the full thickness of the skin but not reaching the tendon or bone. Exclusion criteria were: allergy to antibiotics used in solution for AM decontamination, transcutaneous oximetry value below 30 mmHg for patients with diabetes mellitus, known history of AIDS or HIV, ankle-brachial index (ABI) < 0.6 for all patients except those with diabetes mellitus, suspicious for cancer or history of radiation at the wound site, severe (uncontrolled) systemic disease, or planned surgical intervention.

Before enrolment in the study, all patients were pretreated by SOC for a minimum of six weeks in centers. In case the patients reacted positively, i.e., the wound closure was more than 20% of the original wound area, they continued to be treated in a standard way and were not included in the study. Patients with wound closure inferior to 20% after six weeks of SOC pretreatment (42) were enrolled in the study based on the results of the examination (inclusion and exclusion criteria) after signing the informed consent. They were alternately assigned to the SOC (29) or AM group (14), whereas the other 21 patients in which the treatment in the SOC group was ineffective were transferred to the AM group (Figure 1).

### 2.3. Patients

In total, 42 Caucasian patients, 27 men, and 15 women, of average age 65 ± 14 (26–85 years), with a total of 54 NHW, were enrolled in the study. Thirty-seven wounds (68%) were venous, six (11%) arterial, six (11%) traumatic, three (6%) postoperative, one (2%) pressure, and one (2%) of diabetic origin.

Patients’ medical histories included hypertension (76%), hyperlipidemia (38%), diabetes mellitus II (26%), chronic venous insufficiency (19%), peripheral artery disease (17%), chronic atrial fibrillation (21%), ischemic heart disease (5%), chronic renal disease (2%), chronic heart failure (2%), or anemia (7%) without the primary etiology of NHW. Also, 76% of patients had at least two comorbidities, 55% of patients suffered from three, and 21% of patients from five. None of the patients suffered from systemic skin or subcutaneous tissue disorder or autoimmune disease. None of the diabetic patients had glycated hemoglobin higher than 9%. The wound resistance to previous treatments before the enrolment into the study spanned from 6 weeks to 27 years (1404 weeks) with an average of 124 weeks. Wound size varied between 0.7 and 70.6 cm^2^, averaging 20 ± 23 cm^2^. For detailed patient demographic data, see Table 1.

In the SOC group, 29 patients (19 men, 10 women) with a total of 36 wounds were involved in the study and treated for at least 6 weeks. The average age of SOC patients was 65 years, the mean time from the onset of the wound to the beginning of therapy was 144 weeks, and the average wound size was 22 cm^2^. Patients not responding to the treatment, showing less than 20% wound closure after 6 weeks of SOC, were transferred to the AM group.

The AM group totaled 35 patients (43 wounds); 21 patients were recruited from the SOC group, and 14 were included directly). The average age of AM patients (23 men, 12 women) was 67 years, the mean length of time from the onset of the wound to the beginning of therapy was 167 weeks, and the average start size of the wound was 16 cm^2^.

### 2.4. Wound Treatment Procedure, Subjective Patients’ Feelings

Patients were followed up every week based on standard visit protocol. Each visit included the wound treatment (wound debridement, a sample for microbiological examination, disinfectant solution application, the application of SOC or AM, and secondary fixation dressing application), an assessment of the patient’s subjective condition, and photo documentation. All relevant data were recorded during the visit in the patient form.

The primary coverage in SOC patients was chosen according to the condition of the wound and the amount of exudate. In general, deeper defects were covered with a combined gel covering containing sodium alginate (NU-GEL™, Systagenix, Wound Management, Gatwick, United Kingdom) or containing hypochlorite and sodium hypochlorite (Granudacyn gel) together with a layer of absorbent covering based on hydrofiber containing silver ions. Shallow defects were covered with a foam cover with a silver-containing silicone layer (Mepilex^®^ Transfer Ag, Mölnlycke Health, Göteborg, Sweden) to maintain negative cultures or reduce the bacterial load.

In AM groups, AM was standardly applied weekly up to week eight and bi-weekly from the ninth week. The application frequency was further adjusted according to the evolution of the healing progress and overall patient status. AM was applied as described previously [33]. Briefly, after the cryopreserved AM allograft was thawed, rinsed with sterile saline, and applied to the cleaned wound with a minimum of 5 mm overlap to ensure complete contact with the wound surface, the graft was fixed with a secondary foam cover (Mepilex XT, Mölnlycke Health, Göteborg, Sweden) with an overlap of at least 2 cm and fixed with a bandage. In patients with venous insufficiency, a compression bandage was added. The wound covering was left for two to five days depending on the defect condition. Patients were provided with the necessary material for home dressings or were assisted by the Home Care Agency.

Subjective pain perception was evaluated using a visual analog score (VAS) on a scale from 0 (no pain) to 10 (the worst pain). Quality of life was assessed using the Questionnaire Quality of Life (QoL) on a scale from 0 (the best QoL) to 68 (the worst QoL) for patients with chronic wounds [9,35].

### 2.5. Healing Evaluation

Each wound was photo-documented with a scale indicating the center, patient ID number, defect number, and the visit date. The data were centralized, and the wound size was determined independently (by two trained persons) by manually tracking the wound border on calibrated images with automatic determination of the area size using NIS-Elements software (Laboratory Imaging, Prague, The Czech Republic).

Based on the final status of the treatment, wounds were divided into three groups: healed (wound closure of 99–100%), partially healed (wound closure of 50–99%), and unhealed (wound closure of 0–50% or wound area increase compared to the baseline) [33,36].

### 2.6. Statistics

Photographs of wounds and patient forms (pain, QoL) from all centers were collected and analyzed in anonymized forms. An independent statistician (MB) analyzed the data sets. The visual analog score and Wound-QoL score data were evaluated via a standard linear mixed effects model [37,38] with two factors (final status and week) and random patient effect in order to reflect autocorrelation in data measured on the same individual. Pairwise comparisons among time points were corrected for multiple comparisons via Tukey’s HSD procedure [39]. Further, the time to partial wound closure was analyzed by the Kaplan–Meier approach [40,41]. The survival curve estimates were converted to cumulative distribution function for easier reading. The mean wound healing trajectory was smoothed with a GAMM (Generalized Additive Mixed Model, Wood 2017 [42], with random individual-specific effects) using complexity-penalized splines with penalty coefficients estimated from data via generalized cross-validation [43]. The statistical computations were performed in R Core Team [44] using mgcv and survival packages.

## 3. Results

In the SOC group (36 defects), complete healing was achieved in 7 wounds (19.4%), partial healing in 1 patient (2.8%, 1 wound), and 28 wounds (77.8%) remained unhealed(Table 2). The period required for the defect’s complete healing in the SOC group lasted from 3 to 126 weeks in healed (mean 30 weeks), 87 weeks in the partially healed group (one patient only), and the treatment varied from 6 to 14 weeks (mean 8 weeks) in the unhealed group. After 6 weeks of SOC treatment, the wound area was calculated, and patients with wounds <20% closure were transferred to the AM group.

From the AM group (43 defects), 24 (55.8%) defects responded to the AM application by complete healing, 10 (23.3%) wounds healed partially, and 9 defects (20.9%) were resistant to the treatment. Complete wound closure was reached within 4 to 65 weeks (mean 30 weeks, 4–43 AM applications were needed), partially healed patients were treated for 13 to 87 weeks (mean 55 weeks, 14–68 AM applications), and unhealed patients for 5 to 55 weeks (mean 23 weeks, 6–45 AM applications). Of 28 wounds resistant to SOC treatment, which were consequently treated by AM allografts, 14 were healed completely. The data summarizing outcomes for SOC and AM treatment are presented in Table 2, and the study outcomes based on the wound’s origin are presented in Table 3. Wound healing progress (wound closure) of all defects is presented in Figure 2.

In cases where the wound healed quickly (healed subgroups from both SOC and AM groups), relatively rapid granulation and epithelialization were observed, followed by rapid closure of a larger area of the wound (progressive phase of healing). After that, in the range from ten to twelve weeks of application of AM allograft, healing proceeded more slowly (slower phase of healing) (Figure 2A,E).

Wound closure in SOC and AM-partially healed groups progressed slowly without reaching complete closure despite receiving the longest care periods. The wound closure reached 80.1% (one patient only) and 70.3% for SOC and AM groups, respectively (Figure 2B,F).

Wounds from unhealed subgroups in SOC and AM patients did not respond to treatment despite intense care (Figure 2C,G). The wound closure values oscillated around zero, meaning their size repeatedly changed from positive (healing) to negative effect (wound area enlargement compared to the baseline). The statistical estimates of average wound closure with a confidence interval (CI) of 95% for all three subgroups are shown in Figure 2D,H for SOC and AM-treated defects. The data show that the healing progress with SOC and AM is very similar. The worsening in UH patients after AM application after 30 weeks of treatment is due to the deterioration of the two wounds that were the only ones monitored at that time (P28D2, P41D1).

We conducted a Kaplan–Meier based-analysis to compare the healing of AM and SOC groups. For 25% closure, the analysis showed that after 25 days of treatment, the probability of wound closure was significantly higher for AM compared to SOC. For 50% closure, there were no statistically significant differences between both groups (Figure 3).

The assessment of pain in the AM group shows that the mean pain score declined significantly from baseline (2.9) to 1.8, 0.8, and 0.24 after the first, fifth, and tenth weeks of AM treatment, respectively (Figure 4). A significant drop from baseline was found between the first and tenth weeks of application in healed and partially healed patients with finally unhealed defects. The difference between the baseline and last week was 2.6, which was highly significant (*p*-value < 0.001 ***) for all wounds. The estimated mean score in the last week was 0.72 (95% confidence interval was 0.41; 1.03). While the pain relief gradually progressed up to the tenth week of treatment for healed and partially healed groups, a stagnation between weeks five and ten was observed for the unhealed group.

Similarly, the quality of life assessed from the QoL form show a gradual decrease of negatively perceived factors from baseline (week 0) to week 1, 2, 5, and 12, respectively (Figure 5). The difference between the QoL baseline and the last week was 15.65, which was highly significant (*p*-value < 0.001 ***) for all wounds. The estimated mean score in the last week was 11.35, and the 95% confidence interval was 7.69; 15.38.

No adverse secondary reaction related to the AM allograft application was observed.

Twenty-five out of 35 patients treated with AM were followed (71%), i.e., 32 wounds (74%). Of the healed wounds, 71% remained healed, and 12.5% (3 wounds) relapsed. The follow-up period lasted an average of 30 months (4–66 months). Of the wounds that did not heal (groups PH, UH), only one wound (5.3%) was healed in the follow-up period (P29D1).

## 4. Discussion

In this multicenter study, we analyzed the effect of cryopreserved AM allograft applications on the healing of NHW. In a group of 42 patients with 54 nonhealing wounds of various origins that lasted, on average, for 31 months, we showed that the mean healing capacity of SOC treatment reached approximately 21%, while the application of cryopreserved AM reached about 69%.

This study shows that the application of AM is also effective and suitable for the healing of polymorbid elderly patients, whose wound healing is more complicated due to numerous alterations from homeostasis, which interfere with the healing process [45]. For patients suffering from two comorbidities, 63% were successfully healed; for patients with three or four comorbidities, 42% were healed; and for patients with five to eight comorbidities, 63% were healed completely. It reflects that polymorbidity does not affect the effect of AM application for NHW healing, importantly when compared with patients suffering from a single pathology only. These results are surprising and may be partially explained by the complex care in the hospital centers.

Various factors can influence the efficiency of AM treatment. Among the most pronounced are the method of AM preparation and storage, the wound size, its etiology, time from onset, health status of the subject (polymorbidity, diabetes mellitus, body mass index, age, etc.), and last but not least the AM application frequency [46,47]. Also, it was documented that the individual sensitivity to the AM but not the inter or intra-placental differences are predominant in the efficiency of AM treatment [34].

When comparing our results to other studies that used cryopreserved AM to treat chronic wounds, we are at a similar level of success, which lies between 53–62% [26,33,46,48]. However, the duration of therapy in our study is longer compared to frequently reported ones, which is usually 12–25 weeks [29,33]. Studies using AM for wound treatment often terminate the therapy at the 12th week [26,48,49], so this period is frequently given as a reference time point for treatment efficacy. However, generally around the 12th week of treatment, the healing transitions from the fast-progressive phase to the much slower healing phase occurs. This phase can be longer than the progressive phase [33,50]; see also Figure 2E. Twelve-week periods also coincide with the time limit of most studies evaluating the effect of various skin substitutes on wound healing [36].

In our study, we have chosen long-term healing for several reasons. Our primary goal was to completely heal the patients. The secondary goal was to determine whether the guideline for predicting the wound healing outcome (healed, partially healed, and unhealed) based on wound healing progress between weeks 0–12 [33] is valid for other sets of patients. Of the wounds which healed by 50% or more after the 12th week of AM application, 88% healed completely, and of the wounds which healed by 60% or more, complete closure occurred in 91%. Correspondingly, of the defects which did not even reach 40% closure after 12 weeks, 55% remained partially healed, and 45% stayed unhealed. Wounds that healed completely reached at least 40% closure at the 12th week of treatment. For the group of wounds that do not heal even partially, the oscillation of the size around the baseline is typical for the entire period of 12 weeks of healing. Based on these data, we can confirm a prediction interval for assessing the ability of the wound to heal using AM at 10–12 weeks.

The fact that the application of amniotic or amniochorionic membranes leads to a significant reduction in pain has been repeatedly noted in clinical studies using cryopreserved [31,51,52] or air-dried terminally sterilized tissue [53]. The analgesic effect of cryopreserved AM was mainly explained by the reduction of inflammation and thus secondary to pain relief [31,54]. Recently endogenous bioactive lipids with significant analgesic and nociceptive effects were detected in both placental membranes prepared for grafting [25,55]. It is clear that also anti-inflammatory components present in AM indirectly contribute to the analgesic effect of AM. [21]. In the presented study, we demonstrated that significant pain relief after AM application is not only associated with healing wounds but also with wounds resistant to closure. Some patients from partially healed and unhealed groups requested the continuation of the application of AM allografts for its important pain relief effect, although its wound closure was not significant.

The reduction of pain is also reflected in the improvement of values in the QoL questionnaire. We consider the absence of pain and QoL assessment in the SOC group as a shortcoming of this study. On the other hand, it is known that significant pain relief does not occur after the application of SOC treatment in cases when SOC is not efficient.

We also evaluated the cost of AM treatment, an essential factor in the health care system. For subjects that healed completely, the average cost of the AM membranes used for the wound dressing was approximately 6500 €. For partially healed subjects, the cost was significantly superior (17,000 €) due to our selected strategy of no time limit on the treatment duration until the reaction to the treatment could be clearly recognized (tendency to heal or not). This is also reflected by the average treatment duration of about 30 weeks for healed patients and 55 weeks for partially healed patients. In the case of unhealed patients, the cost came to 8300 € with an average duration of 23 weeks.

These data clearly show that: (i) the non-reacting patients could be rapidly identified, and the AM application should be aborted and replaced by an alternative treatment, (ii) the well-responding subjects could also be identified in the early stages of the treatment, and the cost of AM application is favorable compared to the inefficient SOC, which in case of some subjects included in the study lasted several years while the AM treatment resulted in the complete healing within several weeks. Therefore, the most problematic group are the patients reacting only weakly but positively to the AM application. It is difficult to identify whether the subject will finally heal in the long term, which implies an elevated cost of treatment, or whether only an incomplete but significant level of wound closure will be achieved. However, it should also be emphasized that also for partially healed patients, the AM treatment resulted in an important improved quality of life. On the other hand, we also observed and confirmed during the study that most of the wounds that healed only partially after AM treatment (PH subgroup) degraded progressively after the end of the therapy, which was also associated with the degradation of QoL and pain perception. In summary, this suggests that recognizing the subjects with a strong potential to heal as early as possible after the AM treatment has been started can significantly improve the economic aspect of the AM treatment, which is undoubtedly more expensive than SOC.

The efficiency of the AM application is also supported by the fact that out of 25 wounds treated by SOC in the centers but not reacting at all to the therapy, 14 reached complete healing after AM application, and 5 exhibited partial healing. Six others did not react to the AM application suggesting that in some particular subjects, the chance for healing is very limited, probably even with an alternative modern approach.

## 5. Conclusions

Our results show that AM application to long-lasting NHW is beneficial, even in polymorbid elderly patients. While high-quality and intensive SOC in the centers resulted only in 19% of subjects reaching complete healing, the AM application raised this level to about 56%. In polymorbid patients, 42–63% of wounds were healed based on the number of comorbidities per patient. The results also show that AM treatment cannot be considered a universal solution for all individuals as, probably due to the physiological and somatic differences, some subjects do not respond or respond only partially to AM application. Therefore, the early distinction (after 12 weeks of AM application) into responding, partially responding, and non-responding wounds is essential for the cost-effectiveness of AM application on nonhealing wounds.

## Figures and Tables

**Figure 1 bioengineering-10-00900-f001:**
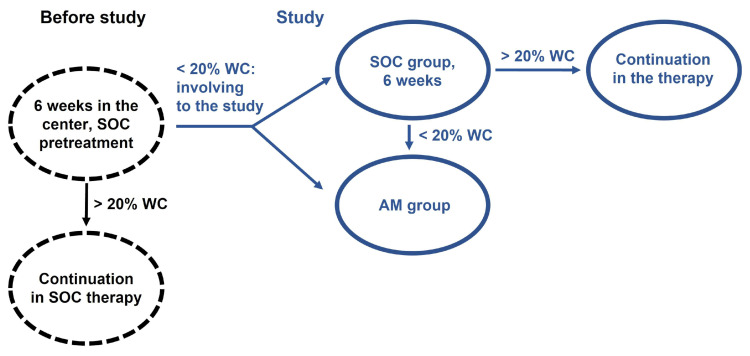
Schema of the patient’s enrolment in the study. Before the study, a six-week pretreatment using SOC was performed. If the wound closure (WC) was larger than 20% compared to the baseline wound area, the treatment using SOC continued (black). If the wound closure was less than 20%, the patient was included in the study (blue), alternately in the AM or SOC group. After six weeks of SOC treatment, patients with WC > 20% were continued in the SOC group. Non-healing patients (WC < 20%) were relocated to the AM group.

**Figure 2 bioengineering-10-00900-f002:**
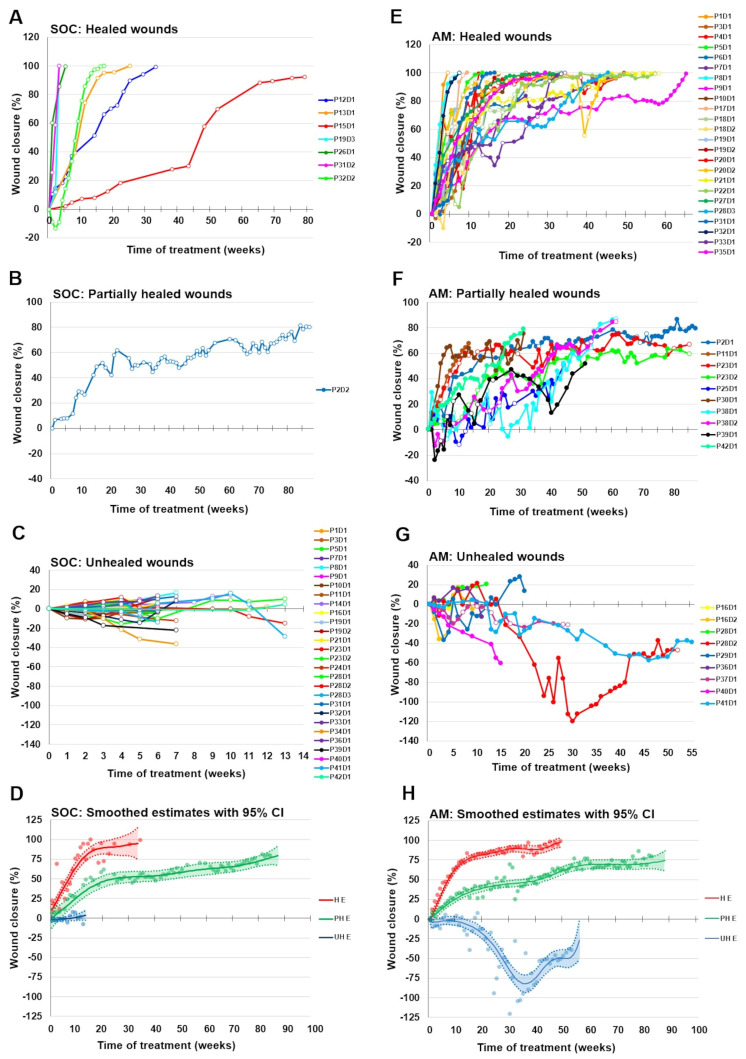
Wound healing dynamics of all assessed defects in SOC (**A**–**D**) and AM allografts (**E**–**H**). Wound closure is presented separately for healed wounds (**A**,**E**), partially healed (**B**,**F**) wounds, and unhealed defects (**C**,**G**). (**D**,**H**) Smoothed weekly averages (solid lines) with 95% confidence intervals (dotted lines) compared with empirical weekly averages (dots). The different final status groups: healed (**H**), partially healed (PH), and unhealed (UH) defects treated with SOC (**D**) and AM (**H**), are distinguished by different colors. Due to the significant late onset of healing and a long period to complete closure (126 weeks, not shown on the graph), the data from defect P15D1 (see part A, red line) were not included in the calculation of smoothed estimates.

**Figure 3 bioengineering-10-00900-f003:**
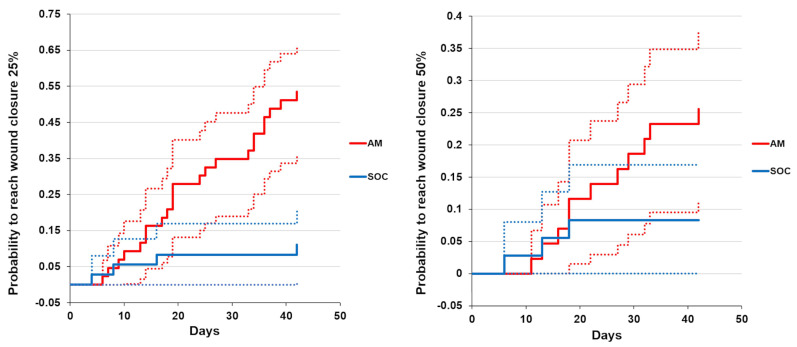
Kaplan–Maier-based analysis of partial wound healing time distribution (25% and 50% wound closure). Comparing probability cumulative distribution functions (CDF) for amniotic membrane (AM) and standard of treatment (SOC) groups. Estimates of CDF are plotted as solid lines. Their 95% confidence interval limits are plotted by dotted lines.

**Figure 4 bioengineering-10-00900-f004:**
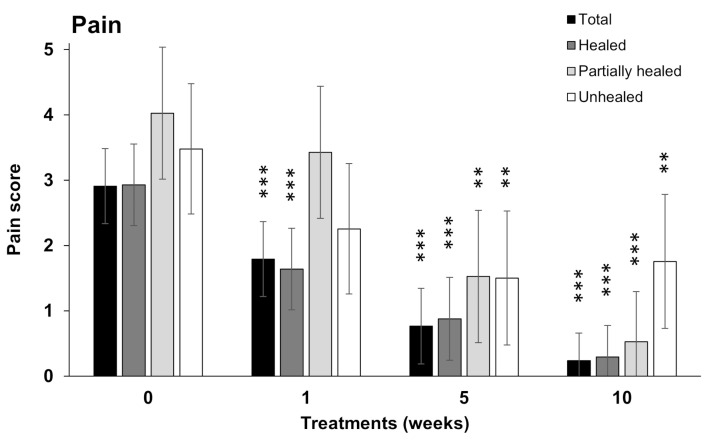
Pain score after the application of AM allografts. Estimated mean score ± 95% confidence interval from all patients on a scale from 0 (no pain) to 10 (the worst pain) at week (W) 0, 1, 5, and 10 of treatment. The data are presented separately for all (total), healed, partially healed, and unhealed defects. ** *p* < 0.01, *** *p* < 0.001.

**Figure 5 bioengineering-10-00900-f005:**
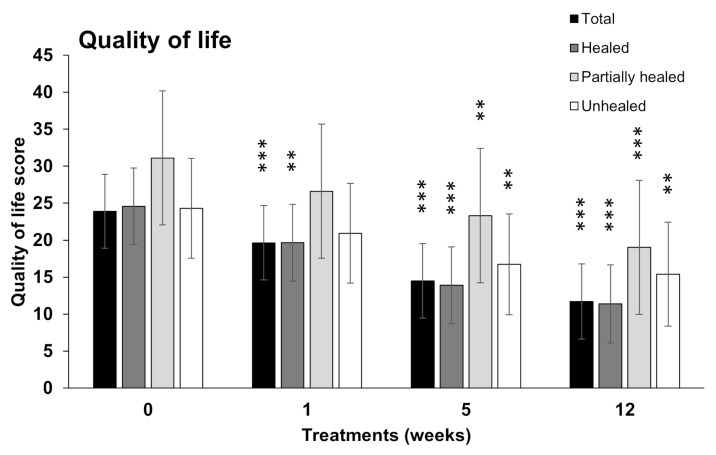
Quality of life score (QoL) after the application of AM allografts. Estimated mean QoL score ± 95% confidence interval at week (W) 0, 1, 5, and 12 of treatment. The data are presented separately for all (total), healed, partially healed, and unhealed defects. ** *p* < 0.01, *** *p* < 0.001.

**Table 1 bioengineering-10-00900-t001:** Patients’ demographic data and baseline characteristics.

Patients	All Patients	SOC Patients	AM Patients
Number of Patients	42 (100%)	29 (100%)	35 (100%)
Age, in years (mean ± SD)	65 ± 14	65 ± 16	67 ± 12
Age ≥ 65 years: N, (%)	28 (67)	21 (72)	25 (71)
Male (N, %)	27 (64)	19 (66)	23 (66)
Smoker (N, %)	7 (17)	5 (17)	6 (17)
BMI (mean ± SD)	29 ± 5	29 ± 5	29 ± 5
Obese BMI ≥ 30 (N, %)	13 (31)	8 (28)	10 (29)
Diabetes mellitus II (N, %)	11 (26)	9 (31)	10 (29)
HbA1c (mean % ± SD)	6.1 ± 1.1	6.2 ± 1.2	6.1 ± 1.1
Albumin (g/L) (mean ± SD)	40 ± 5	40 ± 5	40 ± 5
Albumin ≤ 35 g/L (N, %)	8 (19)	5 (17)	7 (20)
Total protein (g/L) (mean ± SD)	72 ± 6	72 ± 5	72 ± 6
ABI 0.6–0.9 (N, %)	12 (29)	7 (24)	11 (31)
ABI > 0.9 (N, %)	15 (36)	11 (38)	11 (31)
Comorbidities			
Two and more chronic diseases (N, %)	32 (76)	22 (76)	28 (80)
Three and more chronic diseases (N, %)	23 (55)	16 (55)	20 (58)
Five and more chronic diseases (N, %)	9 (21)	6 (21)	8 (23)
**Wounds**	**All Defects**	**SOC Defects**	**AM Defects**
**Number of Wounds**	**54 (100%)**	**36 (100%)**	**43 (100%)**
Start size (mean ± SD, cm^2^)	20 ± 23	22 ± 23	16 ± 19
Time from onset (weeks, mean ± SD)	124 ± 219	144 ± 257	167 ± 290
Time from onset >1 year (weeks)	21 (39)	16 (44)	18 (42)
Type			
Venous (N, %)	37 (68)	23 (63)	32 (74)
Arterial (N, %)	6 (11)	5 (14)	2 (5)
Traumatic (N, %)	6 (11)	5 (14)	5 (12)
Postoperative (N, %)	3 (6)	1 (3)	2 (5)
Pressure (N, %)	1 (2)	1 (3)	1 (2)
Diabetic (N, %)	1 (2)	1 (3)	1 (2)
Location			
Calf/Midfoot (N, %)	29 (54)	20 (55)	22 (51)
Forefoot (N, %)	5 (9)	5 (14)	3 (7)
Hindfoot/ankle (N, %)	17 (31)	9 (25)	16 (38)
Toe (N, %)	1 (2)	1 (3)	1 (2)
Other (N, %)	2 (4)	1 (3)	1 (2)

ABI = ankle-brachial index, BMI = body mass index, HbA1c = glycated hemoglobin, N = absolute number, SD = standard deviation.

**Table 2 bioengineering-10-00900-t002:** Study outcomes in the standard of care treatment (SOC) and amniotic membrane (AM) groups. Wound closure progress is indicated for the SOC or AM treatment, separately for SOC/AM-healed (SOC-H, AM-H), SOC/AM-partially healed (SOC-PH, AM-PH), and SOC/AM-unhealed (SOC-UH, AM-UH) wounds.

Wound Treatment	SOC	SOC-H	SOC-PH	SOC-UH	AM	AM-H	AM-PH	AM-UH
No of wounds: N (%)	36 (100)	7 (19.4)	1 (2.8)	28 (77.8)	43 (100)	24 (55.8)	10 (23.3)	9 (20.9)
Start size (mean ± SD, cm^2^)	22 ± 23	23 ± 30	87 ± NA	20 ± 18	16 ± 19	15 ± 15	20 ± 29	15 ± 15
Start size median	12.2	10	87	12.2	11.2	8	10.3	8.6
End size (mean ± SD, cm^2^)	16 ± 18	0 ± 0	17.4	20 ± 18	5 ± 11	0 ± 0	6 ± 7	17 ± 17
End size median	9.2	0	17.4	11.7	10.7	0	3.4	9.7
Wound closure area (%)	21	100	80.1	−0.8	68.5	100	70.3	−16.6
Treatment duration (mean ± SD, weeks)	14 ± 24	30 ± 44	87	8 ± 3	34 ± 23	30 ± 18	55 ± 26	23 ± 18
Treatment duration (median, weeks)	7	17	87	7	31	30	56	15
Mean N of AM applications (from-to)	NA	NA	NA	NA	24 (4–68)	20 (4–43)	38 (14–68)	18 (4–45)
Number of visits	9	11	74	6	29	26	44	19

N = absolute number, NA = not applicable, SD = standard deviation.

**Table 3 bioengineering-10-00900-t003:** Study outcomes based on wound origin. The number (N) and percentage (%) of wounds in standard of care treatment (SOC) and amniotic membrane (AM) groups are shown. Wound closure progress is indicated separately for SOC-healed/partially healed/unhealed (SOC-H/SOC-PH/SOC-UH) and for AM-healed/partially healed/unhealed (AM-H/AM-PH/AM-UH) wounds.

Wound Type	SOC	SOC-H	SOC-PH	SOC-UH	AM	AM-H	AM-PH	AM-UH
Venous (N, %)	23 (100)	2 (9)	1 (4)	20 (87)	32 (100)	16 (50)	8 (25)	8 (25)
Arterial (N, %)	5 (100)	3 (60)	0 (0)	2 (40)	2 (100)	2 (100)	0 (0)	0 (0)
Traumatic (N, %)	5 (100)	1 (20)	0 (0)	4 (80)	5 (100)	4 (80)	0 (0)	1 (20)
Postoperative (N, %)	1 (100)	1 (100)	0 (0)	0 (0)	2 (100)	1 (50)	1 (50)	0 (0)
Pressure (N, %)	1 (100)	0 (0)	0 (0)	1 (100)	1 (100)	0 (0)	1 (100)	0 (0)
Diabetic (N, %)	1 (100)	0 (0)	0 (0)	1 (100)	1 (100)	1 (100)	0 (0)	0 (0)

## Data Availability

Data is contained within the article.

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
