# Peer review of "Outcome of Application of Cryopreserved Amniotic Membrane Grafts in the Treatment of Chronic Nonhealing Wounds of Different Origins in Polymorbid Patients: A Prospective Multicenter Study"

_bioengineering, 2023, doi:10.3390/bioengineering10080900_

Round 1
Reviewer 1 Report
I congratulate the authors on their comprehensive and well-conducted study on the efficacy of deep-frozen human amniotic membrane in healing chronic wounds. Although it seems that a not insignificant part of the data has already been published in Svobodova, A., Horvath, V., Smeringaiova, I., Cabral, J. V., Zemlickova, M., Fiala, R., et al. (2022). The healing dynamics of non-healing wounds using cryo-preserved amniotic membrane. Int Wound J 19, 1243–1252. doi: 10.1111/iwj.13719., the compilation here compared to a new group (SOC) nevertheless contributes to further improved understanding of the application of hAM in wound healing. Overall, the paper is excellently written, yet I would ask for some minor changes before the manuscript can be published.
I am aware that the term cryopreservation is commonly used for freezing. However, this does not correspond to the actual definition. Therefore, in the present work, not cryopreserved hAM has been used, but only deep frozen. (Cryopreservation, by definition, is controlled freezing (meaning controlling of the freezing rate) using CPA to at least -140°C.)
However, in the present work, the membrane was frozen uncontrolled only up to -80°C. I therefore ask you to replace the term cryopreserved by deep frozen throughout the paper, also in the title.
A few more small things:
Page 3, line 124: The sentence structure is confusing, please rephrase, also page 11, lines 338/339.
In general, please explain abbreviations somewhere, even if they seem familiar, e.g. Table 1 N= absolute number?
Table 2 NA = not applicable?
Page 6, line 200: the abbreviation of Generalized Additive Model is GAM
And Table 2, line: Mean No of AM applications (from-to) What is meant here by from-to, if only a single value is listed?
The conclusion of the work that certain wounds do not respond to hAM treatment and thus no longer need costly further treatment with it is valuable. Have the authors investigated in what ways the non-responsive wounds differ from the healing ones? Underlying disease? Prior duration of existing wound? Age of the patient?
Or perhaps such an investigation is planned?
Reviewer 2 Report
The study concerns the use of cryopreserved amniotic allografts in a treatment of non-healing wounds of various origin. There are several major and few minor comments listed below:
1. The term “non-healing wound” seems to be misleading, especially since Authors have shown that, despite this name, even in SOC group some of wounds healed completely, some partially, and only part of them really deserves that name. Possibly, rather “hard-to-heal” would be more adequate…
2. The most difficult to follow is the number of patients and wounds being analyzed. Authors have stated that the study involved 42 patients with 54 wounds. Then, they randomly (???) assigned 29 patients to SOC, 21 to AM and another 14 patients to AM. It gives total 64 patients. The wounds calculation is similar – 36 in the SOC and 43 in AM gives 79 wounds instead of 54. Also, what does “randomly” mean – alternately, using any allocation scheme, or just by doctors’ opinion?
3. The main and primary sin of this report is the composition of study group. Wounds of various origin were mixed – and this was not good for the clearness of analysis. The patients were allocated “randomly” and the distribution of wound types is not equal – and this is very bad, since these groups were different by definition. In case of venous ulcers, which are usually easier to treat, Authors had 74% wound allocated to AM group versus 63% in SOC. On the other hand, ischemic ulcers, usually non-treatable without causative treatment (i.e. limb revascularization) were more frequent in SOC 14% vs 5% in AM group.
4. The next issue is the comparison of various wounds using the pain assessment. In diabetic foot ulcers, especially in neuropatic patients, usually the pain assessment is useless due to decreased pain sensation. On the other hand, in the ischemic ulcer the problem is not only the wound pain, but the limb pain in general due to critical ischemia. Finally, in patients with venous ulcer the proper use of compression results in the reduction of venous hypertension and tissue edema and has the best analgesic effect.
5. Authors should compare the healing rate rather by wound type (main disease), than healing efficacy. It should not be surprising that in “healed wounds” group the healing time is much shorter than in “unhealed wounds” group…
6. The beneficial effect of amnion dressing on wound healing has several components and this part of discussion should be more extensive. Apart from endogenous bioactive lipids amniotic membrane contains various anti-inflammatory and pro-angiogenic factors including cytokines (VEGF, ANG, EGF, TGF-b) and protease inhibitors (TIMPs, serpins, etc.). Moreover, it also consists of enormous amounts of hyaluronic acid with strong pro-regenerative and anti-inflammatory properties – compare series of papers by Litwiniuk et al., e.g. J Wound Care - doi: 10.12968/jowc.2017.26.8.498., Wound Repair Regen. doi: 10.1111/wrr.12188, or, particularly, Wounds. 2016 Mar;28(3):78-88.
7. What is the benefit to use amniotic membrane in case of polymorbid patients? While this is emphasized in the title of manuscript, it is surprising that Authors do not include any comment regarding this issue in final conclusions. Or there is no particular difference?
Minor comments:
1. In Table 1. ABI 0.07 - Please verify, since according to exclusion criteria (line 114) patients with ABI <0.6 should not be included. Also, ABI≥0.90% seems to be a typo error. Please, verify.
2. Fig. 1, especially graphs C, E and F with their extensive legends are completely unreadable.
Round 2
Reviewer 2 Report
No further comments. Thank you